# Online patient feedback as a measure of quality in primary care: a multimethod study using correlation and qualitative analysis

Anne-Marie Boylan ![ORCID], Amadea Turk, Michelle Helena van Velthoven ![ORCID], John Powell

Nuffield Department of Primary Care Health Sciences, University of Oxford, Oxford, UK

**Correspondence to**
Dr Anne-Marie Boylan; anne-marie.boylan@phc.ox.ac.uk

## ABSTRACT

**Objectives** To ascertain the relationship between online patient feedback and the General Practice Patient Survey (GPPS) and the Friends and Family Test (FFT). To consider the potential benefit it may add by describing the content of public reviews found on NHS Choices for all general practices in one Clinical Commissioning Group in England.

**Design** Multimethod study using correlation and thematic analysis.

**Setting** 1396 public online reviews and ratings on NHS Choices for all General Practices (n=70) in Oxfordshire Clinical Commissioning Group in England.

**Results** Significant moderate correlations were found between the online patient feedback and the GPPS and the FFT. Three themes were developed through the qualitative analysis: (1) online feedback largely provides positive reinforcement for practice staff; (2) online feedback is used as a platform for suggesting service organisation and delivery improvements; (3) online feedback can be a source of insight into patients' expectations of care. These themes illustrate the wide range of topics commented on by patients, including their medical care, relationships with various members of staff, practice facilities, amenities and services in primary care settings.

**Conclusions** This multimethod study demonstrates that online feedback found on NHS Choices is significantly correlated with established measures of quality in primary care. This suggests it has a potential use in understanding patient experience and satisfaction, and a potential use in quality improvement and patient safety. The qualitative analysis shows that this form of feedback contains helpful information about patients' experiences of general practice that provide insight into issues of quality and patient safety relevant to primary care. Health providers should offer patients multiple ways of offering feedback, including online, and should have systems in place to respond to and act on this feedback.

## INTRODUCTION

Patient experience is a core component of quality healthcare. Recent high-profile inquiries into care failures in the English NHS have uncovered a failure to take account of patients' concerns.[1–3] These inquiries have called for all organisations to solicit the

---

### Strengths and limitations of this study

► A multimethod approach combining both quantitative and qualitative approaches allows us to investigate the value of online patient feedback as both a quantifiable measure of quality (eg, through correlation with other scores) and to explore the content and draw conclusions about how people use reviews.

► The online reviews lack demographic data, so it is difficult to draw conclusions about the characteristics of people who post.

► The General Practice Patient Survey and Friends and Family Test have their own weaknesses, and it is therefore debatable whether they represent a gold standard with which to correlate.

---

experiences of patients and carers, recognising their experiences as essential to monitoring quality and safety in the NHS. In the NHS, patient experience and satisfaction is recorded in primary care using the General Practice Patient Survey (GPPS) and the Friends and Family Test (FFT). However, the FFT has been criticised for its invitation and response biases, and because it has resulted in a significant amount of staff time spent collecting, collating and reporting on the data, rather than devoting this time to quality improvement.[4] The usefulness of the GPPS is also debatable and has been criticised as items pertaining to the patient-doctor relationship are reported at practice level, potentially masking individual general practitioner (GP) performance. A study in English general practices found that positive survey responses can mask negative experiences that patients described in subsequent interviews[5] and equally prevents GPs from reflecting on their practice.[6] This suggests that surveys might not capture a full and holistic picture of patients' experiences and

**BMJ**

that providing a platform on which patients can describe their experiences in an unstructured way may counteract this problem. In fact, there is no gold standard measure of patient satisfaction and experience in primary care and in this context, online patient feedback websites may offer a solution.

Online patient feedback is becoming increasingly prevalent.[7] A recent UK survey showed that 42% of respondents had read and 8% had posted online feedback about healthcare experiences on various types of patient feedback websites.[8] Early evidence indicated some correlation with standardised measures of patient satisfaction in secondary care with online feedback about secondary care.[9] It may provide an efficient and effective means of collecting information about patient experience and satisfaction, not necessarily replacing current standardised measures, but offering a way to complement their content. The emergence of online feedback is also seen as potentially useful in monitoring and inspection[10]—in 2013, the Care Quality Commission invited websites that collect patient feedback to share data for use in their monitoring activities. At the same time, GPs express a range of concerns about online patient feedback, particularly in relation to its usability, validity and transparency.[11] Equally, patients in general have mixed views about the appropriateness of posting reviews online. A qualitative interview study showed that it can be a convenient way of publicly sharing feedback, but that patients are concerned about accessibility, privacy and security, and about how seriously doctors would take it.[12]

In addition to the concerns of GPs and patients, there are other factors in general practice that may complicate the reception and use of online feedback. General practice provides a different context for online reviews and ratings than secondary care. The smaller nature of each organisation means that there is greater potential for staff and patients to be identifiable in reviews. Unlike most secondary care organisations, general practices do not tend to have dedicated patient experience managers or communications staff, and the resource (finance and time) implications of reading and responding to feedback may often be prohibitive.

In this context, we undertook a multimethod study to examine the relationship between the content of online patient feedback on the NHS's patient feedback website, NHS Choices and standardised measures of patient experience and satisfaction (the GPPS and FFT), acknowledging that these measures are not without their flaws. Our aim was to determine if there was a correlation between online reviews and ratings (both qualitative and quantitative feedback) and other quality measures. We also aimed to identify what the content of online reviews reveals about patient experience and satisfaction with general practice, and if it has the potential to provide additional benefit to understanding experiences of primary care.

## METHODS
### Study design
This is a multimethod study of online patient qualitative reviews and quantitative ratings for each general practice in Oxfordshire Clinical Commissioning Group (CCG) in England. A multimethod approach, combining both quantitative and qualitative approaches, allows us to investigate the value of online patient feedback data as both a quantifiable measure of quality, including through correlation with other frequently used measures, as well as to explore content and draw conclusions about the usability of reviews. Other measures include the FFT, which asks patients 'How likely are you to recommend our service to friends and family if they needed similar care or treatment?'.

### Setting
This study was conducted on all general practices in Oxfordshire CCG in England, which, at the time of data collection, included 70 general practices, serving approximately 700 000 registered patients. Oxfordshire CCG covers a mixed rural/urban population which is relatively affluent although there are pockets of deprivation with significantly poorer outcomes in terms of health, education, income and employment. In 2018, 87.4% patients reported having a positive experience of their GP practice compared with a national (England) average of 83.8%; and the total percentage of Quality Outcomes Framework points obtained across Oxfordshire CCG was 97.6% compared with an England average of 96.3%.

Data for each practice were extracted from NHS Choices, the GPPS and the FFT. More information on the general practices can be found at oxfordshireccg.nhs.uk. NHS Choices (http://www.nhs.uk) is the UK's biggest health website, containing a range of information about health conditions and health services. In addition to learning about the staff and facilities at any general practice, patients can post reviews and ratings of their experiences of using a general practice to the NHS Choices site. Patients enter their feedback (reviews and ratings) on a page dedicated to their general practice. There are some instructions provided on how to do this. All reviews are anonymised by NHS Choices before they are publicly available using specific moderation rules, which include removing other names, including staff names, and swear words. No identifiable information is published. Online patient feedback lacks accompanying demographic data, so conclusions about the characteristics of those who post are not possible. General practice staff can access these comments and can respond online if they choose.

### Data sources
All patient reviews and ratings for each general practice in the Oxfordshire CCG posted from October 2009 to July 2016 were extracted from NHS Choices in October 2017. The reviews were in text format and the ratings were numeric, on a scale of 1–5 stars. The GPPS and the FFT data were downloaded from gp-patient.co.uk and

england.nhs.uk/fft (the NHS England websites), respectively, for July 2016. The total proportions of respondents with a good experience (very and fairly good) for the 'Overall experience of GP surgery' and 'Recommending GP surgery to someone who has just moved to the local area' scores were extracted from the GPPS. The total proportion of respondents recommending the practice (extremely likely and likely) for the 'likelihood to recommend the practice to friends and family' score from the FFT were extracted for each practice in the CCG.

### Quantitative analysis of the reviews and ratings

Quantitative analyses were conducted using SPSS V.22. Each of the 70 practices was given a unique identifying number. Reviews were checked by two researchers and duplicates removed. Descriptive analyses were conducted to demonstrate the trend in frequency of reviews, and the proportion of positive, negative and mixed comments. We report the median and IQR for the number of reviews. We used Spearman's Rho to determine correlations between positive and negative reviews, the GPPS and the FFT and report the Spearman correlation, $R^2$ and p value (p<0.05 considered to be significant). The content of the qualitative reviews were assigned a numeric value to categorise them as either entirely positive (1) or entirely negative (0). These comments contained either only positive or only negative items. Mixed responses, that is, containing both positive and negative items were also categorised and assigned a numeric value.[2] The proportions of positive, negative and mixed responses reviews were calculated by dividing the number of those reviews by the total number of reviews. For the GPPS, the total proportions of respondents with a good experience were calculated by combining the proportions of respondents who had a 'very good' or 'fairly good' experience and respondents recommending the practice were calculated by combining the proportions of respondents who said 'definitely' or 'probably' recommend the practice to someone who had just moved to the local area. For the FFT, the total proportions were calculated by combining the proportions of respondents who were 'extremely likely' or 'likely' to recommend the practice to friends and family. Using Spearman's Rho, the proportion of positive and negative comments were individually compared with the GPPS 'good experience' and 'likely to recommend the practice' scores and the FFT 'likely to recommend the practice' score. The NHS Choices reviews were compared with their accompanied star ratings to research whether the valence of reviews matched their star ratings (eg, whether negative reviews had low star ratings, mixed reviews had medium star ratings and positive reviews had high star ratings).

### Qualitative analysis of the reviews

We adopted an inductive thematic approach[13] to analyse the qualitative reviews. This allowed us to explore and search for patterns in the subjective experiences reported in the reviews. The reviews were analysed by the first and second author (AMB and AT) and NVivo 11 was used to aid the data management process. A coding frame was developed inductively in discussion with the research team and was updated when new codes were added. The emergent findings were discussed in regular meetings. The resulting themes were developed inductively and in discussion with the wider research team. The qualitative analysis was conducted before the quantitative analysis in an attempt to ensure that it was not influenced by the quantitative findings.

To ensure quality, we drew on Yardley's[14] principles of good qualitative research. To demonstrate *sensitivity to context*, we drew on a comprehensive scoping review of relevant literature to inform this research and obtained ethical approval from the University of Oxford (reference R53128/RE001) prior to commencing the research. Skilled and experienced researchers undertook thorough data collection and in-depth analyses to demonstrate *commitment and rigour*. We kept a clear audit trail and used appropriate methods, demonstrating *transparency and coherence* and we consider the *impact and importance* of this work in the discussion below.

### Patient and public involvement

Patients and members of the public were not involved in planning or conducting this study. However, they were consulted about a wider programme of work on online feedback and agreed that exploring the content of patient feedback for primary care was an important project.

## RESULTS

### Quantitative analysis of the reviews and ratings

At the time of data collection (October 2016), there were 1402 reviews in total for the 70 practices. Six were verbatim repetitions (ie, posts by the same users at the same time) indicating they were errors and so were excluded from further analyses, leaving a total number of 1396 included reviews. Every general practice in this CCG had received at least one review on NHS Choices. The median number of reviews was 17 (IQR: 9–28). One surgery had received only one review and the highest number of reviews received by any surgery was 142. The earliest was recorded on 13 October 2009. Of the 1396 reviews, 59% (n=823) were positive, 34% (n=474) were negative and the remainder 7% were mixed (n=99).

### Correlation with FFT

Our correlation analyses showed that practices with a larger proportion of positive reviews had a significantly higher FFT score (Spearman correlation=0.595, $R^2$=0.299, p=0.000) and those with a larger proportion of negative reviews had a significantly lower FFT score (Spearman correlation=−0.625, $R^2$=0.333, p=0.000) (see table 1 and figure 1).

**Table 1** Correlation (Spearman) between the proportion of positive responses and Friends and Family Test score (those who would probably or definitely recommend the practice)

| | Proportion positive vs Friends and Family Test score | Proportion negative vs Friends and Family Test score |
|---|---|---|
| Correlation coefficient | 0.595* | −0.625* |
| Significance (2-tailed) | 0.000 | 0.000 |
| Total number | 70 | 70 |

*Correlation is significant at the 0.01 level (2-tailed).

**Table 2** Correlation between the proportion of positive response and proportion of GPPS respondents with an overall positive experience (very good or fairly good)

| | Proportion positive vs GPPS positive overall experience | Proportion negative vs GPPS positive overall experience |
|---|---|---|
| Correlation coefficient | 0.527* | −0.560* |
| Significance | 0.000 | 0.000 |
| Total number | 70 | 70 |

*Correlation is significant at the 0.01 level (2-tailed).
GPPS, General Practice Patient Survey.

### Correlation with GPPS

General practices with a larger proportion of positive reviews had a significantly higher proportion of positive GPPS comments (Spearman correlation=0.527, $R^2$=0.279, p=0.000). General practices with a larger proportion of negative reviews had a significantly lower proportion of positive GPPS comments (Spearman correlation=−0.560, $R^2$=0.315, p=0.000) (see table 2 and figure 2).

General practices with a larger proportion of positive reviews had a significantly higher proportion of patients from the GPPS survey recommending the surgery (Spearman correlation=0.595, $R^2$=0.279, p=0.000). General practices with a larger proportion of negative reviews had a significantly lower proportion of positive GPPS comments (Spearman correlation=−0.625, $R^2$=0.334, p=0.000) (see table 3 and figure 3).

Eighty per cent (n=1117) of the 1396 reviews were accompanied by a star rating, of which 44% (n=600) had received a five-star rating, the highest possible score. 28% (n=307) had received the lowest rating of one. The

spread of star rating scores is shown in figure 4, clearly demonstrating a U-shaped distribution.

### Ratings versus reviews

Of the 307 one-star ratings, 96% (n=294) were accompanied by a negative review. Of the 600 five-star reviews, 96% (n=578) were accompanied by a positive review. Of the 55 three-star ratings, 58% (n=32) were negative, 35% (n=19) positive and the remainder were mixed (table 4).

### Qualitative analysis of the reviews

In this section, we present the findings of the qualitative analyses of the comments on the general practices. Three themes were developed through an iterative process and in discussion with the research team: (1) online feedback largely provides positive reinforcement for practice staff; (2) online feedback is used as a platform for suggesting service organisation and delivery improvements; (3) online feedback can be a source of insight into patients'

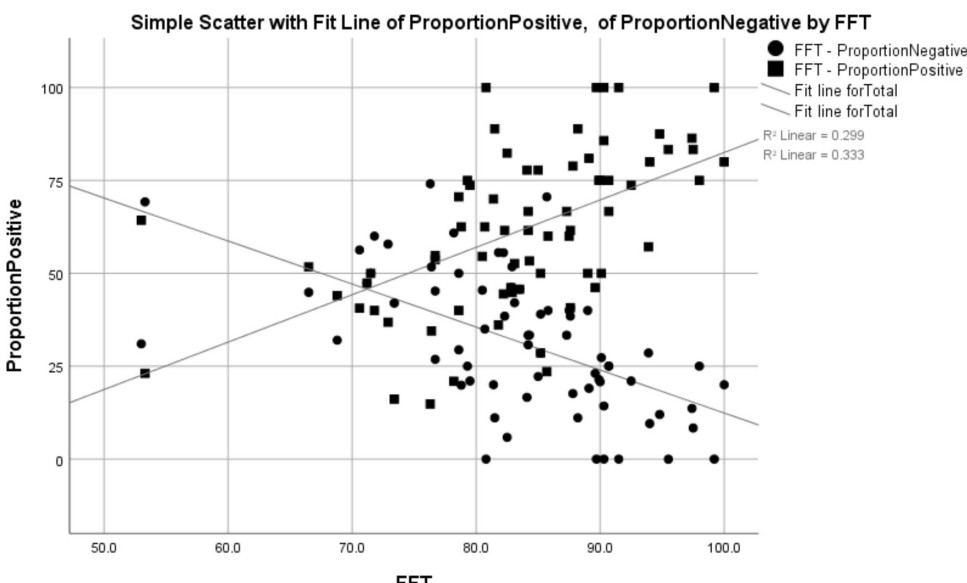

**Figure 1** Correlation between the proportion of positive responses and the FFT score (those who would recommend the general practice). FFT, Friends and Family Test.

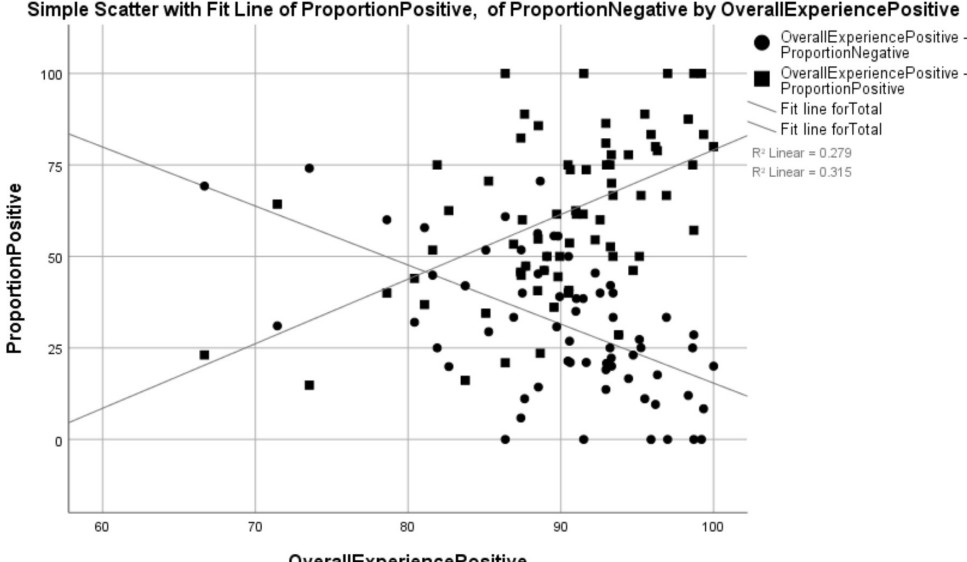

**Figure 2** Correlation between the proportion of positive responses and proportion of GPPS respondents reporting an overall positive experience. GPPS, General Practice Patient Survey.

expectations of care. Sample quotations from the reviews illustrating the themes are found in box 1.

The comments were about the full range of practice staff, including GPs, practices nurses, midwives, receptionists and pharmacists. Reviews about GPs, nurses and midwives frequently, but not exclusively, recounted positive experiences of care. Reviews about receptionists often included negative content. Practice managers were not explicitly mentioned in the comments. Patients often used reviews to express satisfaction or dissatisfaction with their interactions with staff. They also used them as an opportunity to express gratitude for the care they received.

### Online feedback largely provides positive reinforcement for practice staff

Reviews were largely positive and reviewers sometimes prefaced or concluded their positive comments with how they were surprised at the negative reviews and ratings their practice had received. In response to this, they often included a defence of the practice in the positive report of their care experiences. This demonstrates the positive esteem in which patients who comment online hold their general practices and that they wanted staff to know they supported them and felt positively about their care experiences.

Reviewers made largely favourable comparisons between their current practice and others they had previously attended or had heard about. Patients' comparisons with other practices and with doctors within practices were sometimes unfavourable, but for the most part were positive. The reviews also contained comparisons of doctors within each practice, demonstrating that patients drew on previous experiences in writing their reviews and not necessarily on one single interaction. These findings suggest that those who provide online patient feedback draw on their personal histories and relationships with the practices and practice staff when reviewing their experiences (see box 1).

### Online feedback is used as a platform for suggesting service organisation and delivery improvements

The comments frequently referred to the services offered by the practice, how the patients experience them and the way in which services were organised and delivered. They discussed a range of service delivery issues. These included access and appointments, which were largely a source of frustration for patients, who frequently acknowledged that GPs did their best to work within the strict time constraints they were under. Time taken to get an appointment was frequently reported in the reviews. This related to the time spent trying to get through on the phone and talking to the receptionists in addition to the delay in availability of appointments with patients citing waiting times of 3 weeks and longer. Opening hours was also another contentious access issue. Practices were

**Table 3** Correlation between the proportion of positive responses and proportion of GPPS respondents who recommended the surgery

| | Proportion positive vs GPPS recommending the practice | Proportion negative vs GPPS recommending the practice |
|---|---|---|
| Correlation coefficient | 0.595* | −0.625* |
| Significance | 0.000 | 0.000 |
| Total number | 70 | 70 |

*Correlation is significant at the 0.01 level (2-tailed).
GPPS, General Practice Patient Survey.

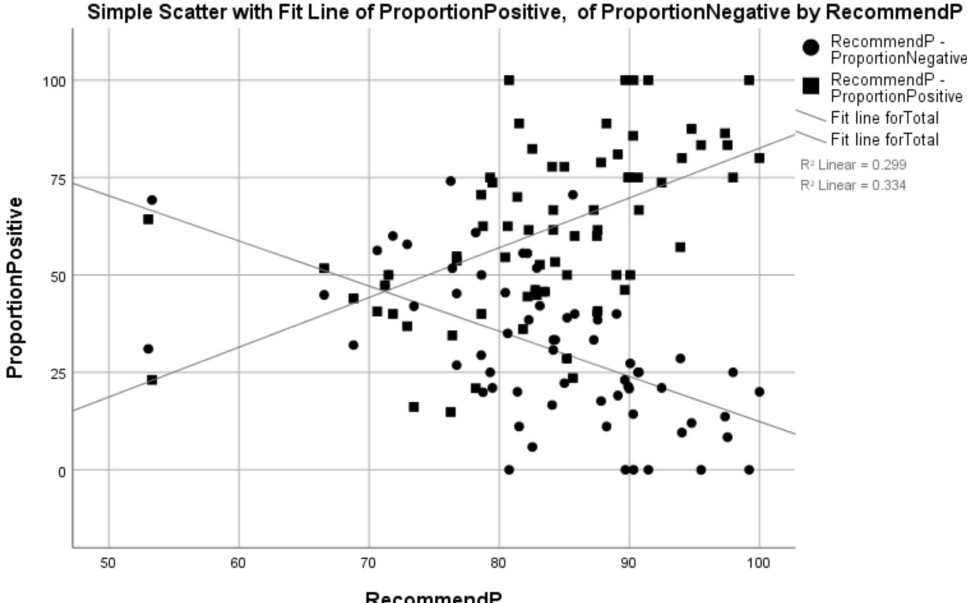

**Figure 3** Correlation between the proportion of positive responses and proportion of GPPS respondents who recommend the general practice. GPPS, General Practice Patient Survey.

criticised for closing for lunch and others were praised for offering appointments in the evenings and on Saturdays. Continuity of care was often discussed alongside the issue of appropriate provision of staff. Many comments referred to not being able to see their named GP or to see the same GP twice about the same issue. This was not a concern for all, as comments stated GPs took the time to review their medical notes.

Other services that were commented on, included automated check-in machines, booking systems and online services, which again received mixed feedback. How these improved efficiency for patients or made attending appointments more complex was explored in the reviews. Telephone access and triage were again mixed with patients particularly commenting on the role of receptionists as gatekeepers. There were concerns about receptionists asking about the reason for the call without having any medical training. Comments also considered the physical environment and focused on the building, particularly its accessibility, aesthetics and cleanliness.

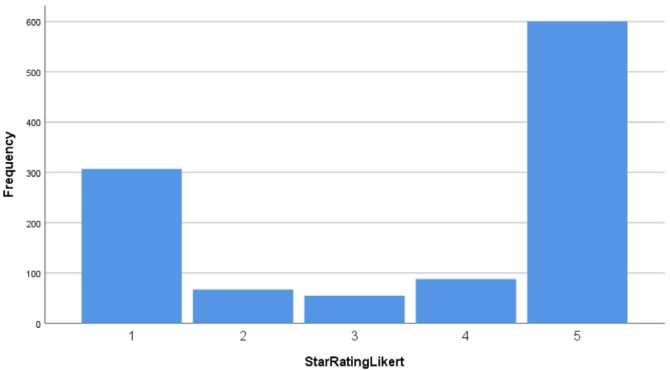

**Figure 4** Proportion of star ratings received across all general practices.

Facilities, mainly parking and waiting rooms, were subject to review with concerns on the latter pertaining particularly to issues of privacy. Comments included those posted by patients who had a substantial history with the practice and contained information about how they perceived the practices had changed over time, offering insight into how they had improved or declined.

Patients' feedback often included suggested improvements, indicating that providing online feedback was not used simply as a chance to complain or moan about their practice. The range of items contained in the feedback in this theme demonstrates the broad range of issues that are important to patients' experiences of attending their general practice and it is clear that they feel confident in reviewing aspects their experience that relate to service delivery and organisation.

### Online feedback can be a source of insight into patients' expectations of care

Positive accounts of care by all healthcare staff often included adjectives such as 'personal', 'compassionate' and 'respectful'. Descriptions of positive care often centred on communication skills with reviewers commenting on times when healthcare staff listened and took the time needed to explain the diagnosis, treatment, side effects and what to expect next. They also included accounts of shared decision-making and involvement in decisions about care.

Poor communication was also reported frequently in the negative or mixed reviews. This tended to be about not being listened to, feeling rushed and treated with suspicion, particularly with regard to medication requests or repeat prescriptions. There were multiple negative evaluations of the doctors' attitudes, with reviewers using words like 'rude', 'unfortunate manner', 'unfriendly' 'dismissive', 'hostile', 'condescending' and 'disinterested'.

**Table 4** Star ratings vs reviews

| Star rating | | Negative review | Positive review | Mixed review | Total reviews |
|---|---|---|---|---|---|
| 1 | Number | 294 | 3 | 10 | 307 |
| | % within star rating | 95.8% | 1.0% | 3.3% | 100.0% |
| 2 | Number | 60 | 3 | 4 | 67 |
| | % within star rating | 89.6% | 4.5% | 6.0% | 100.0% |
| 3 | Number | 32 | 4 | 19 | 55 |
| | % within star rating | 58.2% | 7.3% | 34.5% | 100.0% |
| 4 | Number | 10 | 60 | 18 | 88 |
| | % within star rating | 11.4% | 68.2% | 20.5% | 100.0% |
| 5 | Number | 8 | 578 | 14 | 600 |
| | % within star rating | 1.3% | 96.3% | 2.3% | 100.0% |
| Total | Number | 404 | 648 | 65 | 1117 |
| | % within star rating | 36.2% | 58.0% | 5.8% | 100.0% |

The reviews contained accounts of instances when patients' dignity and privacy were compromised by the action or inaction of staff, providing insight into how staff could improve the patients' experiences by prioritising dignity and privacy. Reviewers also commented on their perceptions of the competence of the staff they encountered. They recounted experiences that led them to feel like they could not trust their healthcare practitioner's advice. Comments also addressed misdiagnoses, feeling dismissed, queries around staff competence and suspicions around mistakes. These included global statements, like 'the GP misdiagnosed me on several occasions' that contained no specific information to assess the veracity or gravity of the concern. However, comments also contained specific detail about medical aspects of their condition (see box 1). Comments containing specific information may indicate the level of engagement some patients have with their care and possibly their expectations of how GPs should therefore interact with them. It was also notable that patients were aware of the constraints GPs were working within and recognised that they could not always give the care they wanted to.

Reviewers also commented on their interactions with receptionists using both positive and negative terms: 'exceptional', 'helpful', 'efficient', 'kind and respectful' and 'rude', 'brusque', 'didn't listen', 'incompetent'. These extremes demonstrate the variation of experiences that are reported online. Criticisms of and praise for receptionists often pertained to their manner and not their efficiency or competence. They were often described as 'customer-facing' or performing a 'customer service' role. This perception conflicted with some aspects of their role, including the questions they asked when patients phoned to ask for an appointment or to speak to the doctor. What is clear is that receptionists are seen as the face of the practice and can influence how patients feel about the care they receive from booking and checking in for appointments to the interaction with their GP or the practice nurse.

This theme provides an understanding of patients' expectations of care and interactions with general practice staff, which often centred around the level of interaction they expected with their GP. A sense of thoroughness and completeness was important in interactions with GPs, while good 'customer care' was often cited in relation to interactions with receptionists.

## DISCUSSION

This study found a relationship between online patient feedback and other quality measures, specifically the GPPS and the FFT, in general practices in one English CCG. We found a moderate positive correlation between the online feedback on NHS Choices and both of these quality measures. Online patient feedback was found to express the extremes of experience, the very positive and the very negative, as demonstrated by the U-shaped distribution of the frequencies of different ratings (figure 4). The majority of the ratings were positive with few middle-ground experiences being reported. This also suggests that it is not appropriate to take an arithmetic average (mean) score from these data, given the skewed distribution towards each end of the spectrum.

Through our qualitative analysis, we developed three themes that indicate how patients who post online feedback review their experiences. We demonstrated that they use NHS Choices to provide positive reinforcement for staff, to suggest improvements to service delivery and organisation, and we found that the comments contained a great deal of information about their expectations of care. Reviewers commented on almost the full range of practice staff; only practice managers were omitted from the reviews, perhaps because they are less likely to interact with patients than other staff. The vast majority of comments were positive and pertained to a range of factors about the care experience, including the environment, service delivery and interactions with staff. However, this analysis demonstrates that patients also

## Box 1 Sample quotations from the online reviews

**Online feedback largely provides positive reinforcement for practice staff**

*I am surprised at some of the adverse comments about this GP surgery. I have been registered with the practice … for many years and would be devastated if I had to change surgery; my GP has always been excellent, listens and is proactive in referring me on for other services if required.*

*I transferred to this practice from a different surgery […] because it was almost impossible to get an appointment with my previous GP. This surgery is so much better the availability of appointments with doctors and nurses is great. … my experience is that they are much better than other GPs in the area.*

**Online feedback is used as a platform for suggesting service organisation and delivery improvements**

*[…] Getting appointments can be a challenge! I have been a patient of this surgery for 30 years and things are a lot better now than they were 15 years ago!*

*The appointment side of things is also ridiculous. The earliest appointment I've managed to get recently has been 3 weeks in advance! Which when you need to see someone fairly urgently isn't acceptable.*

*Very long wait each time. The worst was today where I had to wait for 1.5 hours to see the GP despite arriving punctual for my appointment.*

*The doctor I am listed with is excellent, I have been less impressed when I have had to see another doctor in the practice, who is much less approachable and seemed rather dismissive.*

*I am very unhappy with the disabled parking at this medical centre. I am a wheelchair user and I can not use the one space they have. Its too small…*

*I like the new TV and music in the waiting room, it gives a more relaxed feel and something to pass the time with some interesting information*

*…Could also do with some new chairs in the waiting room to match their new extension and to keep patients comfortable while they wait!*

*Premises are cramped and overcrowded. Playing Radio 2 very loud in the waiting room "for reasons of confidentially" is not great if you have a headache or are feeling unwell.*

*Sometimes it can be difficult to get through on the phone but an extended surgery until 8pm one1 day a week is very useful for me*

*An overcrowded surgery, with too many parttime GPs. This means that it is difficult to see the same GP each time and there is therefore no real continuity of care.*

*I have never seen the same doctor more than once; however, I have no complaints about all the doctors whom I have seen.*

**Online feedback can be a source of insight into patients' expectations of care**

*Always on one's level with superb listening skills and adequately explaining things. Never any sense of rush. I always feel at ease and appropriate nice humour shared is good. So very polite and they say they are sorry to have kept me waiting.*

*One of the practice nurses is also excellent, capable of carrying out the most intimate of procedures without causing embarrassment.*

*There are three doctors in the surgery. I have found all of them well-informed and caring, taking time to explain procedures, results, and options.*

*I put three stars for involvement in decisions because sometimes the doctors themselves have no choice or they have very less alternatives for treatment or for referral options, so they are limited to help with a range of options. This limitation is mostly due to the system itself*

## Box 1 Continued

*within which they need to work in. But all staff try and help as much as possible.*

*They told me to undress, did not offer me a gown to cover myself and made me lay on the table next to an open window with partially open curtains.*

*For COPD they initially prescribed half the normal dose of inhaler, presumably in order to cut costs. They didn't give me any guidance about exacerbations. More recently at my annual review my SpO2 (oxygen saturation) was recorded as 98% rather than the actual value of around 94%. The 98% would have put me outside the recommended range for review for further medication.*

*…there is one receptionist, who is very helpful just like the others, but very rarely smiles when dealing with patients.*

*The receptionists always pretend that they are the doctors and ask lots of questions that they don't have a clue at all, but eventually, I was always told that a doctor needs to call me back again to discuss the problem. Which is waste of time for everyone, because in every case of mine, the doctor will say that I need to see them anyway.*

comment on issues relevant to quality (eg, autonomy, choice, clarity of communication, confidentiality, dignity, prompt attention and quality of basic amenities) and on issues of patient safety (eg, access, skill and competence of clinicians and clinical errors, although examples of these were few).

To our knowledge, no other study has shown a relationship between quality measures and online feedback in primary care. Other studies have found correlations with online feedback in secondary care in England[9] with the inpatient survey and the Hospital Consumer Assessment of Healthcare Providers and Systems in the USA.[15] We acknowledge that the FFT and GPPS are not without their problems, but there is no gold standard measure of quality or safety in primary care with which to compare online feedback. The position of online feedback, therefore, may be to provide supplementary information on issues of patient experience in primary care and pluralise the range of media through which patients can report their experiences.

Our findings are consistent with previous research that has shown that the majority of online feedback is positive.[7 16–18] This is contrary to the opinions of GPs, who have been found to perceive online feedback as predominantly negative.[11] Also consistent with previous research is the U-shaped distribution of the weighting of online feedback, which was reported in a study of a German patient feedback website on which approximately 50% of the feedback was aimed at general practitioners.[19]

Berwick[3] argued that the NHS should be a 'system devoted to continual learning and improvement of patient care' (p. 5). He also called for more transparent reporting on quality and safety data and emphasised the importance of listening to patients and carers. This is particularly important as what constitutes quality or good care may not be consistent across all populations.[20] Online feedback websites may provide a partial solution

to this, offering patients the opportunity to see how others have reviewed their care. Equally, rating and review sites could act as databases of experiential insight, thus potentially useful to healthcare providers aiming to incorporate patients' views in service organisation and delivery.

In addition to quality improvement, online feedback has the potential to improve patient safety.[21] A small minority of patients commented on the medical aspects of their experiences (eg, oxygen saturation levels in chronic obstructive pulmonary disease (COPD)); most only mentioned their condition or disease to explain why they needed to see the doctor. We suggest that patients may have the capacity to comment online on this level and thus on issues pertinent to patient safety. This has been demonstrated in previous studies of patient safety in primary and secondary care.[22 23]

Patients' views on safety in primary care have previously been researched qualitatively and through patient reported experience and outcome measures. Communication has been shown to be crucial in improving patient safety, along with timely access, improved speed of diagnosis and continuity of care.[22 24] Evaluating task performance (the ability of staff to perform particular tasks, largely diagnosis and appreciation of the severity of the problem) was heightened by patients with previous experience of medical harm.[22]

Online feedback should not be the only means of collecting patient insight; a pluralised approach remains warranted.[25] Only a small number of people post reports of their care experiences online. A recent survey in the UK found 8% of respondents had posted feedback,[8] indicating that public awareness is low and perhaps that staff may not encourage this activity. However, the same survey showed that 42% reported reading online feedback, demonstrating the potential power it wields. As we have shown, the majority of feedback is positive and records extremes of experience. GPs tend to perceive online feedback as mostly negative. They may derive more benefit from it if they approached is as capturing extremes of experience that is not representative data where you can take an average, but is a report of individual patient' experiences.

We need to better understand the impact of providing feedback online and to consider the range of possible factors that influence the contents of online reviews. This might include implicit and explicit messages patients receive through how websites like NHS Choices are formatted and through interactions with the health service. Currently, little is known about the difference between providing healthcare feedback via different media and this warrants further exploration.

## Limitations

The feedback data gathered between 2009 and 2016 were extracted from NHS Choices by the research team in 2017 and the correlations with the GPPS and FFT were conducted using the most recently available data, which was from 2016. In addition, and as with previous studies of online patient feedback, we were limited to the information that is available online. Therefore, this study provides little insight into the characteristics of the patients who provide feedback. Equally, NHS Choices moderates online posts by patients and does not publish comments that contravene their rules, including those that are not in English or those containing expletives or staff names. Without access to these unpublished posts, it is unclear if all posts conform to our findings. However, this is the nature of this type of insight and, as such, this study provides a comprehensive analysis of what is available. Adopting a multimethod approach was valuable as it allowed us to correlate the online feedback with established measures of patient satisfaction. Additional insight as to what impact this type of feedback could have was found in the course of the qualitative analysis.

## Future research

As this study has shown, patients comment on a wide range of aspects of their care experience and this insight could be used to make improvements in general practice. However, more research is needed to ascertain whether the findings of this in-depth case study of one CCG could be extrapolated across the NHS to answer the question of whether local insight can be used to make national improvements. National experiences can have local resonance,[26] but it is unclear if the reverse is also true. Online platforms may provide a cost-effective and attractive means for soliciting feedback from patients, but the volume of online reviews per practice is quite low in comparison with the numbers of patients enrolled. Future research should aim to explore the views of service-users who are reluctant to comment online. Equally, we need to explore the views of all staff who are subject to online review, including practice nurses and receptionists, who have been neglected from previous qualitative research in this area. It is unclear how they feel about this phenomenon. In addition, we do not know how online patient feedback is used in primary care and how or if staff can use it to make improvements.[27 28] More research is needed to explore this, particularly how general practice staff perceive and use negative feedback.

## CONCLUSION

Our study shows that patient feedback on general practices found on a national health website is correlated with established measures of patient satisfaction and could be useful in helping patients choose a general practice, in areas where choice is possible. It also shows that it has potential uses in determining issues of quality improvement and patient safety. Health providers should offer patients multiple ways of offering feedback, including online, and should have systems in place to respond to and act on this feedback.

**Acknowledgements** We are grateful to Helen Atherton (University of Warwick) for helpful discussions about this study. We are grateful to the three peer reviewers for their thorough review of the paper.

**Contributors** All authors made substantial contributions to the study design and analysis of the data, were involved in drafting or revising the paper, have approved the final version and agree to be held accountable for all aspects of the work. A-MB designed the study, conducted the literature search, analysed and interpreted the data, drafted the initial manuscript and prepared it for submission. AT contributed to the design of the study, sourced the quantitative and qualitative data and analysed and interpreted the qualitative data and contributed to writing the manuscript. MHvV contributed to the design of the study, prepared the figures, analysed and interpreted the quantitative data and contributed to writing the manuscript. JP contributed to the design of the study, was involved interpreting the data and contributed to writing the manuscript.

**Funding** This research was funded by the National Institute for Health Research (NIHR) Collaboration for Leadership in Applied Health Research and Care Oxford at Oxford Health NHS Foundation Trust.

**Disclaimer** The views expressed are those of the authors and not necessarily those of the NHS, the NIHR or the Department of Health and Social Care. The study sponsors were not involved in the study design; data collection, analysis or interpretation; report writing or decisions about submitting the paper for publication.

**Competing interests** None declared.

**Patient consent for publication** Not required.

**Ethics approval** This study was approved by the Medical Sciences Inter-Divisional Research Ethics Committee, University of Oxford (Ref: R53128/RE001).

**Provenance and peer review** Not commissioned; externally peer reviewed.

**Data availability statement** Data are available in a public, open access repository. All data employed in this paper are available from: http://www.nhs.uk, http://www.england.nhs.uk/fft/friends-and-family-test-data/ and http://www.gp-patient.co.uk.

**ORCID iDs**
Anne-Marie Boylan http://orcid.org/0000-0001-8187-0742
Michelle Helena van Velthoven http://orcid.org/0000-0003-1245-8759

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
