## [Reviewer comments · BMJ Open]

ARTICLE DETAILS

TITLE (PROVISIONAL)	Online patient feedback as a measure of quality in primary care: a multi-method study using correlation and qualitative analysis.
AUTHORS	Boylan, Anne-Marie; Turk, Amadea; Van Velthoven, Michelle Helena; Powell, John

VERSION 1 – REVIEW

REVIEWER	Sara Jackson University of Washington Seattle, WA USA
REVIEW RETURNED	26-Jun-2019

GENERAL COMMENTS	Using online patient feedback to determine quality and patient satisfaction in primary care: a mixed methods case study. General comments: This a mixed methods study of patient comments posted on a national health service website, that includes data for 70 clinical sites. The qualitative work categorized the thematic content of on-line comments, and then the proportion of positive to total comments appears to correlate with survey patient experience data. I think my primary big picture comment is that this correlation between positive comments and positive survey feedback is not surprising. I am also not sure if it provides anything additionally actionable to improve patient care or safety. Maybe clinical sites that are not receiving as much positive feedback could see what patients appreciate about high performing sites? But there are likely to be confounders related to practice characteristics or patient populations serviced that would also be important to understand. I am assuming negative comments are reviewed by the clinical site and action is taken to address confirmed deficiencies, and this would be the most beneficial safety and experience component of the on-line comments (and it sounds like very specific information with names, etc was not available for analysis). - Perhaps add some information on current state use of on-line comments by clinics. Abstract: Theme #2 seems to combine two distinct topics I am guessing that the comments were grouped into positive and negative and that these correlated with FFT and GPPS survey
---

	results – it might help to include more methods in the abstract to clarify this. The conclusion seems intuitive. I suspect a wide variety of feedback, though grouped into themes in this study, would need to be categorized and have actionable items (specific to the feedback) to improve quality of care. If the summary conclusion is that health systems should have mechanisms for patients to give feedback (in addition to more formal feedback survey instruments), and for acting upon the on-line feedback in a fashion that addresses concerns, I could see this. Article summary The third bullet point is not clear to me. It is a limitation that characteristics of patients who provide feedback is not available – is that the point? Introduction Page 3 lines 19-20 This mentions that 42% of patient had “read” health care experiences – can you describe how this is done? Via a patient portal? Do the experiences include doctor’s notes and other parts of their health record? – I see later in the discussion that means read the web comments, could clarify this. After reading the introduction I have a better sense of the context for this study. Perhaps if there is space in the abstract you could add a first sentence in the objective that explains that you are assessing the potentially additive benefit of on-line feedback to established patient experience surveys. Methods Can you explain how and where the patients enter their comments? NHS website, visible to public, any prompt for comments, do they get or expect a response, does the info go to the clinics and do they address deficits, etc. The qualitative and quantitative analytic descriptions are clear and seem appropriate. I do wonder how the mixed reviews were analyzed. If they included both positive and negative components it seems that they should count toward the % positive, unless you are defining this measure as “unequivocally positive” comments (or something like that). It seems possible that a mixed result could have a very strongly positive comment and weak negative, and then this would not be credited to the clinical site. I wonder if you have data that characterizes the clinical sites – such as size, staffing ratios, turn-over of staff, demographic information about patients served such as % receiving government assistance, general health ratings of patient populations, etc, that might confound the relationships between sites and likelihood of positive comments. These would just be helpful potentially in understanding variation among proportion of positive experiences between clinical sites. I appreciate the mention of patient involvement in this patient oriented investigation. Too bad that they were not incorporated and involved in the work. A limitation.
--	---

	Results Page 5 line 20-21 It seems like verbatim repeats should be included if you are quantitatively looking at proportion of positive comments, unless I am misunderstanding something. Over 7 years (or maybe it is one year – 2016?), to have just 17 comments per clinical site seems very low. This makes me wonder about how patients know where to post comments, details of process. Correlation with FFT and GPPS. Perhaps you could describe in more detail these measures. From the methods I thought perhaps you were looking at individual queries from the instruments (like “likelihood for recommending practice”), but if this is a composite score based upon multiple questions this could be made clearer in the methods (and add the surveys in an appendix). Figures: Define GPPS in Figures 2-3 Figure 4 is labels as “proportion” but the graph looks like counts
--	---

REVIEWER	Andrea Hernan Deakin University, Australia
REVIEW RETURNED	10-Jul-2019

GENERAL COMMENTS	Dear authors, This paper addresses a unique topic of online patient feedback association with established satisfaction measures in general practice, as well as analysis of the qualitative online patient comments. The findings show the utility of online patient feedback and the value it may serve to general practice staff for ongoing data monitoring and quality and safety improvement. As the authors state this is the first study to demonstrate such an association and therefore it is important to publish these findings to the wider academic audience. I have some minor comments, as per the attached, that need addressing prior to acceptance for publication. These mainly concern further discussion of some of the findings in relation to the literature, and further detail of the qualitative themes in the results section. Thank you for the opportunity to review this paper, and I wish the authors all the best with the revision and future research in this area. Kind regards, Andrea Hernan The reviewer provided a marked copy with additional comments. Please contact the publisher for full details.
---

REVIEWER	Claire Marsh Bradford Institute for Health Research
REVIEW RETURNED	18-Jul-2019

GENERAL COMMENTS	I was happy to review a paper on this highly topical subject. I think there has been some useful data collected in this research but recommend some very significant revisions to this paper before publication. 1) Substantial further work on the rationale for the two main aims of the paper which I think were a) to correlate online feedback with FFT/GPPS data and b) to analyse the content of online feedback. With respect to a), it is not clear how a correlation with FFT/GPPS would validate the quality of this data source especially as the value of FFT is highly contested, and as the nature of the data is different – qualitative and quantitative. Authors state that one reason is GPSS results are largely positive but mask negative experiences and that online may provide more description to counteract this – does correlation therefore help here? With respect to b), the rationale for why further analyses of the content of online feedback is required is not given. Page 3 lines 37-44 give some of the concerns about online feedback but these are to do with the ‘reception’ and ‘use’ of online feedback – this does not link clearly to what this paper then sets out to analyse which is about content. 2) Some significant methodological issues currently weaken the paper and would need addressing a) quantitative analysis: the correlation of qualitative reviews with survey results was achieved by assigning numerical value of 0 or 1 to either negative or positive online reviews. Some reviews would clearly fit into either category, but an explanation as to what was done with the mixed responses in these correlations is not provided. Rationale for correlating online ratings versus online reviews (Table 4) not provided in the methods section. A summary table of all the numbers of positives and negatives of respective measures would be helpful in following the results section. b) qualitative analysis: whilst methods section describes analysis as an interpretivist, inductive thematic approach, the results outline categories of organised data, not yet developed into themes. The theme headings do not signpost to any interpretation of meaning, but to a collation of data around a topic. E.g., it is not clear what it is about expressing satisfaction or dissatisfaction that is interesting, or indeed whether most express satisfaction or dissatisfaction or a mix of the two. Similarly, what is it about care and communication that is included in the reviews to make this a meaningful theme? I think this then makes the subsequent discussion of qualitative results difficult – page 12 line 3-6 indicate that online feedback is largely positive but I am not sure the qualitative results really support that? As they stand, they just describe the range of responses from positive to negative, not the flavour, and actually the quotes given are mostly negative. I think these problems may all stem from a lack of detail in rationale as to why content of reviews is interesting in the first place. I am sure it is interesting, but these findings need to be located more clearly into academic discourse on this – the authors need to make some decisions on what is interesting here – e.g. is it that the content of online reviews provides something not provided by other data sources? or is it more about the nature of the feedback itself and the types of improvements that therefore need to arise from it?
--

	I also have some specific recommendations:  - Currently I do not think the title reflects the 2 main aims of the paper - P4 line 3 – not a complete sentence - P4 line 44 – are the italics examples of Yardley’s principles? If so, how many principles exist and why are just these ones referred to? I hope that these recommendations help to develop this paper further.
--	--

VERSION 1 – AUTHOR RESPONSE

Reviewer: 1 Reviewer Name: Sara Jackson	
--	--

General comments: This a mixed methods study of patient comments posted on a national health service website, that includes data for 70 clinical sites. The qualitative work categorized the thematic content of on-line comments, and then the proportion of positive to total comments appears to correlate with survey patient experience data. I think my primary big picture comment is that this correlation between positive comments and positive survey feedback is not surprising. I am also not sure if it provides anything additionally actionable to improve patient care or safety. Maybe clinical sites that are not receiving as much positive feedback could see what patients appreciate about high performing sites? But there are likely to be confounders related to practice characteristics or patient populations serviced that would also be important to understand.	Thank you for your helpful review – we have made the amendments you suggest and are grateful to you for helping to improve this paper. Thank you for your comment. We agree that the findings are not necessarily surprising, but we do not consider this as a reason not to publish them, especially as the correlations we conducted have not been done previously.
--	---

I am assuming negative comments are reviewed by the clinical site and action is taken to address confirmed deficiencies, and this would be the most beneficial safety and experience component of the on-line comments (and it sounds like very specific information with names, etc was not available for analysis). - Perhaps add some information on current state use of on-line comments by clinics.	There is not a great deal known about this in practice in primary care and we have added text to say this. We have just completed data collection on a qualitative ethnographic study on the use of online comments in primary care, but we have not finished the analysis, so are unable to report on this. We have included some research on health professional attitudes and have added a line in the discussion regarding learning more about if/how negative comments are perceived and used in general practice.
---	---

Abstract: Theme #2 seems to combine two distinct topics I am guessing that the comments were grouped into positive and negative and that these correlated with FFT and GPPS survey results – it might help to include more methods in the abstract to clarify this. The conclusion seems intuitive. I suspect a wide variety of feedback, though grouped into themes in this study, would need to be categorized and have actionable items (specific to the feedback) to improve quality of care. If the summary conclusion is that health systems should have mechanisms for patients to give feedback (in addition to more formal feedback survey instruments), and for acting upon the on-line feedback in a fashion that addresses concerns, I could see this.	We have made significant amendments to the qualitative analysis and revised the themes accordingly in line with a request from reviewer 3. We have amended the abstract to provide more clarity on the methods, while sticking to the journal's requirements. Thank you – you have helped us clarify our thinking on this. We have amended the conclusion and added similar to the discussion, citing another study we conducted to support this conclusion.
Article summary The third bullet point is not clear to me. It is a limitation that characteristics of patients who provide feedback is not available – is that the point?	Yes, this is correct. We have amended these for clarity.

Introduction Page 3 lines 19-20	
---	--

This mentions that 42% of patient had “read” health care experiences – can you describe how this is done? Via a patient portal? Do the experiences include doctor’s notes and other parts of their health record? – I see later in the discussion that means read the web comments, could clarify this. After reading the introduction I have a better sense of the context for this study. Perhaps if there is space in the abstract you could add a first sentence in the objective that explains that you are assessing the potentially additive benefit of on-line feedback to established patient experience surveys.	We have clarified this. This was based on a previous survey of public awareness and use of online patient feedback sites by Van Velthoven et al. van Velthoven MH, Atherton H, Powell J. A cross sectional survey of the UK public to understand use of online ratings and reviews of health services. Patient Education and Counselling 2018;101:1690–96. doi: 10.1016/j.pec.2018.04.001 Patients’ online comments can be found on structured rating sites, such as NHS Choices, the platform we used in this study. They can also be found on various social media, including Twitter and Facebook. Experiences do not include doctor’s notes or other parts of their health record. We have added this.
--	---

Methods Can you explain how and where the patients enter their comments? NHS website, visible to public, any prompt for comments, do they get or expect a response, does the info go to the clinics and do they address deficits, etc. The qualitative and quantitative analytic descriptions are clear and seem appropriate. I do wonder how the mixed reviews were analyzed. If they included both positive and negative components it seems that they should count toward the % positive, unless you are defining this measure as “unequivocally positive” comments (or something like that). It seems possible that a mixed result could have a very strongly positive comment and weak negative, and then this would not be credited to the clinical site. I wonder if you have data that characterizes the clinical sites – such as size, staffing ratios, turn-over of staff, demographic information about patients served such as	We have added information on this. Thank you. We have clarified that the content of the qualitative reviews were assigned a numeric value to categorise them as either entirely positive (1) or entirely negative (0). These comments contained either only positive or only negative items. The mixed responses were not analysed in these correlations because there were relatively few of them (n=99, 7%) and correlation based on small sample sizes are more likely to be unreliable. In the methods under setting we describe how there are 70 general practices in the CCG and they serve approximately 700,000 patients. We have added
% receiving government assistance, general health ratings of patient populations, etc, that might confound the relationships between sites and likelihood of positive comments. These would just be helpful potentially in understanding variation among proportion of positive experiences between clinical sites. I appreciate the mention of patient involvement in this patient oriented investigation. Too bad that they were not incorporated and involved in the work. A limitation.	additional information about the CCG, including deprivation and patient satisfaction with their general practice in relation to national scores. We agree that this is a limitation.
Results Page 5 line 20-21 It seems like verbatim repeats should be included if you are quantitatively looking at proportion of positive comments, unless I am misunderstanding something. Over 7 years (or maybe it is one year – 2016?), to have just 17 comments per clinical site seems very low. This makes me wonder about how patients know where to post comments, details of process.	Verbatim repeats were posts by the same user at the same time, indicating they were errors, so were removed. The reviews were extracted by the research team in October 2016, but were collected by NHS Choices from 2009 over a seven year period. As we state in the paper, 17 is the median number of reviews, but one general practice received only one and another received 142. So, the number of reviews and ratings of

Correlation with FFT and GPPS. Perhaps you could describe in more detail these measures. From the methods I thought perhaps you were looking at individual queries from the instruments (like “likelihood for recommending practice”), but if this is a composite score based upon multiple questions this could be made clearer in the methods (and add the surveys in an appendix).	general practices in Oxfordshire CCG is indeed low, indicating awareness is low and perhaps that few general practice staff encourage this activity. We have expanded on this in the discussion. We have added more information about the FFT and the GPPS to the methods. We looked at the ‘overall experience’ and ‘likelihood of recommending the practice to others’ scores from the GPPS and the ‘likelihood to recommend the practice to others’ score from the FFT. These are combined proportions. We have added further detail about this in the paper. We have provided the links to the websites where the surveys and data can be accessed.
Figures: Define GPPS in Figures 2-3	We have done this as requested.

Figure 4 is labels as “proportion” but the graph looks like counts	Apologies for this error and thank you for pointing it out, it has been corrected.
Reviewer 2: Andrea Hernan	

Reviewer: 2 **Please see attachment for this reviewer's full review** Reviewer Name: Andrea Hernan Institution and Country: Deakin University, Australia Please state any competing interests or state ‘None declared’: None declared Dear authors, This paper addresses a unique topic of online patient feedback association with established satisfaction measures in general practice, as well as analysis of the qualitative online patient comments. The findings show the utility of online patient feedback and the value it may serve to general practice staff for ongoing data monitoring and quality and safety improvement. As the authors state this is the first study to demonstrate such an	Thank you. Thank you for recognising the new contribution of this work.
--	---

association and therefore it is important to publish these findings to the wider academic audience. I have some minor comments, as per the attached, that need addressing prior to acceptance for publication. These mainly concern further discussion of some of the findings in relation to the literature, and further detail of the qualitative themes in the results section.	Thank you for the helpful literature you suggested. We have incorporated these papers in the discussion and believe it has improved the paper. We are very grateful.
--	--

Thank you for the opportunity to review this paper, and I wish the authors all the best with the revision and future research in this area. Kind regards, Andrea Hernan	Thank you for your thorough and helpful review, which has improved the paper. We have made the amendments you suggested or clarified what we meant as outlined in this table and in the paper.
---	--

Please include a separate ethics statement in the methods section with it's own heading. The 'value' of online reviews might not be necessarily examined via correlations with other measures. Perhaps rephrase this sentence to say you were investigating the association with other measures via correlations. Perhaps rephrase to say 'the quantitative and qualitative nature of the patient feedback data', rather than 'all kinds'. Can the authors please provide more information about the CCG setting? What location in England is it? What types of general practices are included in the CCG - solo clinics, multiple GP clinics? What type of patients are registered with these clinics - what is their socio-economic profile? It is beneficial for the readers to have an idea of the of GP/patient profile and if there were any major differences within the 70 general practices included in the study. It might be worthwhile to include a reference or a link to the NHS Choices website so readers can see what the platform looks like. Was there a specific framework used to assess the qualitative responses as either positive or negative and assign a numerical value? It would be useful to have more information about the process of assigning quantitative values to qualitative data.	We have now included this. Thank you – this is an important point and we agree with you. We have revised this as you suggest. We have amended this accordingly. In the methods under setting we describe how there are 70 general practices in the CCG and they serve approximately 700,000 patients. We have added additional information about the CCG, including deprivation and patient satisfaction with their general practice in relation to national scores. We have now included this We did not use a specific framework. Comments that were entirely positive and contained no negative items were scored as positive; comments that were entirely negative and contained no positive items were scored as negative; comments that contained both positive and negative
--	--

	items were scored as mixed. Positive comments were assigned a numeric value of 1; negative
Tables 1 -3 need some editing to make them readable for publication. They look like an SPSS output at this stage. A table legend is needed to describe the acronyms. I would suggest including Figures 1-4 as supplementary files rather than figures in the paper. The tables and correlation coefficients provide enough data to show the associations. This table (table 4) needs some cleaning for publication. This looks like an SPSS output. I suggest replacing 'count' with 'n'. The first three themes in the qualitative section would benefit from some more detail and explanation like the last two themes in this section. It will give the reader more information about the patient/service user experiences and how they relate to the themes uncovered during analysis. Perhaps rephrase to say 'Quotes illustrating the themes are found in table 5'. Are you also saying here that midwives and receptionists were recounted by patients as 'bad experiences of care'? This statement needs some work as I suspect there may be some nuances to the patients comments. i.e. not all patient experiences of GPs were 'good', and not all patient experiences of receptionists were 'bad'. As above, how was 'good care' assessed by the research team? What literature or framework was used to make this assessment of the patient comments?	comments were assigned a numeric value of 0; mixed comments were assigned a numeric value of 2. We have clarified this in the methods. These have been edited accordingly. Thank you for this suggestion – we have now done this. We have now done this. We have clarified this to ensure the nuance of the patient comments is clear. Thank you – we have made some significant amendments to the qualitative section. Amended as requested We have amended this to provide further clarity on the meaning. By 'good care' we meant positive experiences of care. We have now amended this phrasing accordingly.
This paragraph needs some attention. I'm not sure how 'making comparisons' and 'defending practices' come together to be one theme? They seem like separate themes. Can the authors please provide more concrete	We have now amended these.

examples of how these two categories are linked in one theme. What does this mean? Please elaborate on what review of medical notes means? Were there any patient demographic information collected alongside the online feedback? i.e. gender, age, general practice they attended? This might be useful to know. Perhaps rephrase to 'in one CCG in English general practice'. This currently reads as 'all English general practices'. This paragraph (discussion para 2) would benefit from discussion of the findings in relation to the current literature of patient views on quality and safety in general practice. Are there any similarities or differences between online feedback and research studies? Can you please add a reference here? Fix up reference here Refers to ref 8 in para 3 discussion I'm not sure how this reference matches the statement of the findings. Isn't the U shaped distribution related to extreme star ratings, namely 1 and 5 stars?	We have edited this for clarity: However, the fact that the GP they saw was able to review their medical notes to learn about their medical history meant this was not a concern for all. Patient gender and age are not collected with the patient feedback. In fact, if provided, these details would be removed in moderation. But it is clear which practice they attend and are commenting on. We have added a sentence to clarify this. We have amended this as suggested. We have added some text on this as requested, including some additional references. As requested, we have amended this. Amended We have checked this reference and it supports the finding that of the U shaped distribution to which we refer, which is about the spread of ratings.
“Also consistent with previous research is the U-shaped distribution of the weighting of online feedback, which was reported in a study of a German patient feedback website on which approximately 50 percent of the feedback was aimed at general practitioners.[17]” There is a lot of research demonstrating how patients and carers can comment on patient safety and the contributing factors to patient safety in primary care and other settings. I suggest the authors revise this paragraph to include discussion of their findings in relation to the relevant literature. Please see these references for more information: Hernan A, Giles S, Fuller J, et al. Patient and carer identified factors which contribute to safety incidents in primary care: a qualitative study. BMJ Quality & Safety 2015;24(9):583–93	Thank you for these helpful references, which we have added to the discussion along with providing more discussion on patient safety.

Hernan A, Walker C, Fuller J, et al. Patients' and carers' perceptions of safety in rural general practice. Medical Journal of Australia 2014;201(3 Suppl):S60-S63. Ward JK, Armitage G. Can patients report patient safety incidents in a hospital setting? A systematic review. BMJ Quality & Safety 2012;21(8):685-99 Vincent C, Davis R. Patients and families as safety experts. Canadian Medical Association Journal 2012;184(1):15-16 This last sentence of the paragraph (5) is a separate statement to the preceding information provided. Perhaps elaborate further in a new paragraph or remove the sentence. "This might include implicit and explicit messages patients receive through how websites like NHS Choices are formatted, and through interactions with the health service." Please provide some concrete examples of how this middle ground could be captured? Patient surveys, focus groups, other data collection platforms. "To make feedback websites more effective, we need a better way to encourage people to post middle-ground experiences online."	We have added a new paragraph as you suggest. We have amended this instead to encourage GPs to see this as a report of extremes of experience and that it should be treated as such rather than encouraging middle-ground experiences.
---	--

In the methods it states the data is collection from 2009-2016. Please rephrase this sentence for accuracy. Limitations section Perhaps add some more detail around how patient feedback could be used in general practice for quality and safety improvement.	Thank you for noting this – we have now amended for clarity. We have added information on this as requested, using the helpful references you suggested above.
--	--

Reviewer: 3 Reviewer Name: Claire Marsh Institution and Country: Bradford Institute for Health Research Please state any competing interests or state 'None declared': None declared	
---	--

I was happy to review a paper on this highly topical subject. I think there has been some useful data collected in this research but recommend some very significant revisions to this paper before publication. 1) Substantial further work on the rationale for the two main aims of the paper which I think were a) to correlate online feedback with FFT/GPPS data and b) to analyse the content of online feedback.	Thank you for your thorough review and helpful comments, which we believe have helped improve the paper.
--	--

With respect to a), it is not clear how a correlation with FFT/GPPS would validate the quality of this data source especially as the value of FFT is highly contested, and as	Thank you for raising this. We have added further critique of these measures to deal with this point. The
--	---

the nature of the data is different – qualitative and quantitative. Authors state that one reason is GPSS results are largely positive but mask negative experiences and that online may provide more description to counteract this – does correlation therefore help here? With respect to b), the rationale for why further analyses of the content of online feedback is required is not given. Page 3 lines 37-44 give some of the concerns about online feedback but these are to do with the ‘reception’ and ‘use’ of online feedback – this does not link clearly to what this paper then sets out to analyse which is about content.	aim of the correlation was not to validate, but to compare online feedback with currently used measures to see if it produces similar results, which it does. But also to consider the content of the feedback to see what kind of information it contains and consider its potential to supplement the patient experience data gathered by the GPPS and FFT. We have clarified the rationale for the qualitative analysis, which was to learn what the content of the reviews reveals about patients’ experiences in primary care and to ascertain if there was additional potential benefit provided by the online reviews.
---	---

2) Some significant methodological issues currently weaken the paper and would need addressing a) quantitative analysis: the correlation of qualitative reviews with survey results was achieved by assigning numerical value of 0 or 1 to either negative or positive online reviews. Some reviews would clearly fit into either category, but an explanation as to what was done with the mixed responses in these correlations is not provided. Rationale for correlating online ratings versus online reviews (Table 4) not provided in the methods section. A summary table of all the numbers of positives and negatives of respective measures would be helpful in following the results section.	The mixed responses were not analysed in these correlations because there were relatively few of them (n=99, 7%) and correlation based on small sample sizes are more likely to be unreliable. Table 4 shows what type of review was accompanied by a what star rating. This was done to research whether low star ratings were accompanied by a negative review, medium star ratings by a mixed review and high star ratings by a positive review. We have added this information to the methods section. We believe the other revisions we have made have expanded and clarified the results and discussion and we are also not clear what extra information the reviewer is requesting here. We hope the extensive amendments we have made have provided sufficient clarity.
b) qualitative analysis: whilst methods section describes analysis as an interpretivist, inductive thematic approach, the results outline categories of organised data, not yet developed into themes. The theme headings do not signpost to any interpretation of meaning, but to a collation	Thank you – this was a really helpful comment. We have done quite a lot of work to progress the analysis as you suggest, which has resulted in the development of three themes about the function and nature of

of data around a topic. E.g., it is not clear what it is about expressing satisfaction or dissatisfaction that is interesting, or indeed whether most express satisfaction or dissatisfaction or a mix of the two. Similarly, what is it about care and communication that is included in the reviews to make this a meaningful theme? I think this then makes the subsequent discussion of qualitative results difficult – page 12 line 3-6 indicate that online feedback is largely positive but I am not sure the qualitative results really support that? As they stand, they just describe the range of responses from positive to negative, not the flavour, and actually the quotes given are mostly negative. I think these problems may all stem from a lack of detail in rationale as	online feedback, providing important insights into how patients experience general practice. We have edited table 5 to give more balanced quotations – the number of negative quotes previously given was to give a sense of flavour of the intricacies of the negative experiences that wasn't as necessary with the positive ones. We have also clarified the rationale for the qualitative analysis, which was to learn what the content of the reviews reveals about patients' experiences in primary care and to ascertain if there was additional potential benefit provided by the online reviews.
--	---

to why content of reviews is interesting in the first place. I am sure it is interesting, but these findings need to be located more clearly into academic discourse on this – the authors need to make some decisions on what is interesting here – e.g. is it that the content of online reviews provides something not provided by other data sources? or is it more about the nature of the feedback itself and the types of improvements that therefore need to arise from it?	
I also have some specific recommendations: - Currently I do not think the title reflects the 2 main aims of the paper - P4 line 3 – not a complete sentence	We have amended the title. We have amended this.
- P4 line 44 – are the italics examples of Yardley's principles? If so, how many principles exist and why are just these ones referred to?	There are four. We referred to two because the others are to do with reporting and the impact and importance of the work, which are more for the reader to judge. However, we have included these in the qualitative methods section.
I hope that these recommendations help to develop this paper further.	Many thanks for your thorough review – we appreciate the time you took to help us develop this paper.

VERSION 2 – REVIEW

REVIEWER	Andrea Hernan Deakin University, Australia
REVIEW RETURNED	13-Nov-2019

GENERAL COMMENTS	Dear authors, This paper has been substantially revised and improved according to the previous reviewer comments. There are a few small typos, and a few questions about the qualitative findings that need to be addressed prior to publication. I have used track changes on the attached to highlight where these small changes need to be made. After the authors have made these changes I recommend to accept this paper for publication. Best of luck with your future research. Kind regards, Andrea Hernan The reviewer provided a marked copy with additional comments. Please contact the publisher for full details.
--

REVIEWER	Claire Marsh Bradford Institute for Health Research
REVIEW RETURNED	28-Nov-2019

GENERAL COMMENTS	I really enjoyed reading the revised version of this paper - the flow is much improved and the key messages are much clearer now. I have 2 extremely minor suggestions for revision: 1) The headings for the methods section are slightly misleading. Would they be better represented as something like: 'Qualitative analysis of reviews' and 'Quantitative analysis of reviews and ratings'? If so, the same headings could be used in the results (and possibly in the same order for ease on the reader) 2) In the discussion and/or the section on future research, could the important point about whether or not staff can actually use online feedback for improvement be made? I understand this paper focuses on the value/nature of the feedback collected per se but the wider debate is about whether feedback of any kind can actually be used. There are several papers referring to this problem (e.g. Sheard et al, 2017 The Feedback Response Framework, Soc Sci & Med) but perhaps the most directly relevant is a recently published paper on online feedback responses Ramsey et al 2019 https://pxjournal.org/journal/vol6/iss2/9/). Thanks for submitting a significantly revised version of this paper.
---

VERSION 2 – AUTHOR RESPONSE

We are grateful for your further comments and the small amendments, which we have been happy to undertake, and are very pleased that you are happy with the revised version. Thank you.

Reviewer 2:

We have amended the theme names as you suggested to make them clearer.
We have amended the few changes to the wording as you suggested throughout.

Reviewer 3:

We have renamed the qualitative and quantitative analysis sections as requested in the methods section and the titles are consistent with the findings section.
We have moved the qualitative findings to beneath the quantitative findings as suggested.
We have included references to Sheard and Ramsey's papers as suggested.